# Lightweight Action Recognition in Compressed Videos

Yuqi Huo[1,2], Xiaoli Xu[1,2], Yao Lu[1,2], Yulei Niu[1,2], Mingyu Ding[3], Zhiwu Lu[1,2], Tao Xiang[4], and Ji-rong Wen[1,2]

[1] Gaoling School of Artificial Intelligence, Renmin University of China
[2] Beijing Key Laboratory of Big Data Management and Analysis Methods, Beijing 100872, China
[3] The University of Hong Kong
[4] University of Surrey, UK
luzhiwu@ruc.edu.cn

**Abstract.** Most existing action recognition models are large convolutional neural networks (CNNs) that work only with raw RGB frames as input. However, practical applications require lightweight models that directly process compressed videos. In this work, for the first time, such a model is developed, which is lightweight enough to run in real-time on embedded AI devices (e.g., 40FPS on a Jetson TX2) without sacrifices in recognition accuracy. Compared to existing compressed video action recognition models, it is much more compact and faster thanks to adopting a lightweight CNN backbone. Further, a number of novel components are introduced to improve the effectiveness of the model: (1) A new Aligned Temporal Trilinear Pooling (ATTP) module is formulated to fuse three modalities in a compressed video namely I-frames, motion vectors, and residuals. (2) To remedy the weaker motion vectors (compared to optical flow computed from raw RGB streams) for representing dynamic content, we introduce a temporal fusion method to explicitly induce the temporal context, as well as knowledge distillation from a model trained with optical flows via feature alignment. Importantly, in contrast to existing models that either ignore B-frames or use them incorrectly, our ATTP model employs correct but more complicated B-frame modeling, thus being compatible with a wider range of contemporary codecs. Extensive experiments show that our ATTP outperforms the state-of-the-art alternatives in both efficiency and accuracy.

**Keywords:** Lightweight action recognition, compressed videos, temporal trilinear pooling, knowledge distillation.

## 1 Introduction

Video analysis has drawn great attention from the computer vision community recently due to an increasing demand for automated video content understanding. In particular, videos account for more than 75% of the global IP traffic everyday [11]. With the advancements of deep learning methods, promising

performance has been achieved on a variety of video analysis tasks, including action recognition [19,32,6,37,10,40,8,9,2,29,38,39,42,46,51,7,41], semantic segmentation [24,17], action localization [26,1], and deception detection [5,48].

However, existing convolutional neural network (CNN) based video analysis models still do not meet the requirements of many real-world applications. This is primarily due to two reasons. First, videos such as those on social media sites like YouTube or on smartphones are stored in a compression format to save space. Nevertheless, most existing models work only with uncompressed raw RGB frames. This means that the compressed videos must be decoded first, leading to extra costs on both processing time and storage. Second, most deep video models, based on either two-streams [32,40] or 3D CNNs [37,2,29] are slow (e.g., with high latency for calculating optical flows) and heavy (having a large number of parameters), therefore unsuitable for either processing the vast amount of videos produced everyday, or running on embedded AI devices (e.g., smartphones). Although there are some recent efforts on lightweight video models [3,25], these models do not work on compressed videos.

Recently, researchers start to address the problem of compressed video action recognition [45,30,49,50]. These models take compressed videos (e.g., MPEG-4) directly as input. Such a video contains only a few key frames and their offsets (i.e., motion vectors and residual errors) for storage reduction. However, none of these models is lightweight, and thus they still incur high latency and cannot run on embedded AI devices for edge computing. Apart from the efficiency limitation, existing models are also ineffective due to the fact that they either ignore B-frames or use them incorrectly for extracting motion vectors and residuals. This is despite the fact that in most video bit-streams, most of the frames (more than 60%) are encoded as B-frames to achieve the best compression rate.

In this paper, for the first time, a challenging video analysis task, called *Lightweight Action Recognition in Compressed Videos*, is tackled to fill a gap in video analysis – to the best of our knowledge, this task has not been studied before. The key challenges of lightweight compressed video action recognition are: 1) how to design a lightweight yet highly effective deep CNN model for action recognition; 2) how to extract meaningful representations from compressed videos that contain far less information than the raw RGB frames.

To address these challenges, we propose a lightweight model for compressed video action recognition. Specifically, we adopt EfficientNet [36] as the backbone network to process the multiple modalities extracted from a compressed video (including RGB I-frames $\mathbf{I}$, motion vectors $\mathbf{MV}$, and residuals $\mathbf{R}$). To fuse these modalities, we propose a novel Aligned Temporal Trilinear Pooling (ATTP) scheme to exploit the complementary information contained in them. To remedy the weaker motion vectors (compared to optical flow computed from raw RGB streams) for representing the dynamic content, we introduce a temporal fusion method to induce the temporal context explicitly. Further, since motion vectors are much coarser than optical flow vectors, we adopt a knowledge distillation strategy via feature alignment between our model and a model trained with optical flow extracted from uncompressed videos. Finally, we overcome a limitation

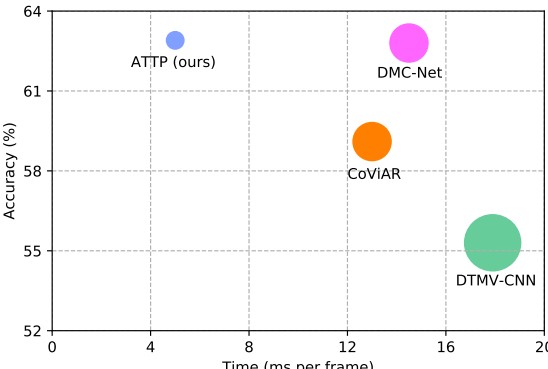

**Fig. 1.** Efficiency and accuracy comparison among various compressed video action recognition methods on the benchmark dataset HMDB-51 [23] with the same platform (i.e., Dell R730). State-of-the-art baselines include DMC-Net [30], CoViAR [45] and DTMV-CNN [50]. The node size denotes the model size (parameters).

of existing models [45,30,49,50] in that they either ignore the B-frames or utilize them incorrectly. This is because computing motion vectors and residuals from the B-frames is far more challenging than from the alternatives (i.e., P-frames). In this work, by employing correct but more complicated B-frame modeling, our proposed model is applicable to all modern video codecs (e.g., H.264 & HEVC). The result of introducing these new components is a lightweight yet powerful action recognition model: it outperforms the state-of-the-art alternatives in both efficiency and accuracy, as shown in Fig. 1.

Our contributions are four-fold: (1) For the first time, we address the challenging problem of lightweight compressed video action recognition. (2) We propose a trilinear pooling module for fusing the multiple modalities extracted from compressed videos. (3) To work with the weak motion vectors, we propose a temporal fusion method to explicitly induce the temporal context, and also boost the backbone trained with motion vectors by distillation with feature alignment. (4) We are the first to exploit both B-frames and P-frames from compressed videos in the correct manner, making our model more compatible with contemporary video codecs. The efficiency test on a fast embedded AI computing device (i.e., Jetson TX2) indicates that our ATTP model can perform video action recognition at about 40FPS (see Table 1). Moreover, as shown in Fig. 1, our ATTP model achieves the best performance but with significantly fewer parameters, as compared to the state-of-the-art methods. This observation is supported by the extensive results reported on two benchmarks widely used for action recognition (see Table 2). To our best knowledge, the proposed model is *the first end-to-end lightweight one* that can perform real-time action recognition on resource-limited devices without sacrifices in recognition accuracy.

## 2   Related Work

**Conventional Video Action Recognition**  Most recent action recognition models [19,32,6,37,10,40,8,9,2,29,38,39,42,46,51,7,41] are based on large CNNs [22]. One of the early models is the two-stream network [32], which is proposed to utilize two CNNs to model raw video frames and optical flow, respectively. Various improved versions [32,40,10,8,9], such as the Temporal Segment Network (TSN) [40], are designed to capture the long-range temporal structure, but they still rely on the optical flow stream, which is expensive to compute. C3D [37] is proposed to model the temporal structure with 3D CNNs. It avoids using optical flow as input. However, it is still much larger than a 2D CNN due to the 3D convolution operations. I3D [2] integrates 3D convolution into a two-stream network and benefits from 2D CNN pre-trained by inflating 2D CNN into a 3D one. One of the key limitations of these models is that they are too heavy for efficient large-scale video analysis. This is particularly true when most videos are stored in compression formats (e.g., MPEG-4 [31] & H.264 [44]), and thus need to be decoded first for these models to run. The recent efforts on lightweight action recognition model design  [3,51,25] only partially address the problem, but still cannot work with compressed videos directly. In contrast, our model is lightweight and also addresses the compressed video action recognition problem.

**Compressed Video Action Recognition**  Due to the limitation of storage space and transfer speed, videos are generally stored and transmitted in a compressed data format. The compression standards, including MPEG-4 [31], H.264/AVC [44], and HEVC [35] (listed in chronological order), commonly use the motion compensation technique that reduces the video data size based on motion estimation from adjacent frames. There are several approaches that leverage useful information from compressed videos for the action recognition task. [18] developed highly efficient video features using motion information based on handcrafted features. DTMC-CNN [49,50] distills the knowledge from optical flows to motion vectors, but the raw video frames are still used for action recognition. CoViAR [45] takes only compressed videos as input, but the whole training process is not end-to-end since multiple modalities are handled separately. Recently, DMC-Net [30] improves CoViAR [45] and achieves state-of-the-art results by adding an optical flow generation network, but both models [45,30] employ the large ResNet-152 [13] as the CNN backbone, which has too many parameters with very high computational cost. Note that both DMC-Net and CoViAR ignore B-frames when extracting motion vectors from compressed videos, while B-frames cover more than 60 percent of total frames in videos. They need to make transformation into old codecs and thus are not compatible with most recent video codecs. Compared to these models, our proposed model is much more lightweight yet more effective in terms of recognition accuracy. It is also more generally applicable by correctly exploiting both B-frames and P-frames. Note that DTMV-CNN [50] also uses B-frames, but in an incorrect way by treating B-frames as P-frames and only considers forward reference. Further, it needs access to both raw and compressed videos to compute motion vectors. In contrast, using compressed videos alone, our ATTP is clearly superior (see Fig. 1).

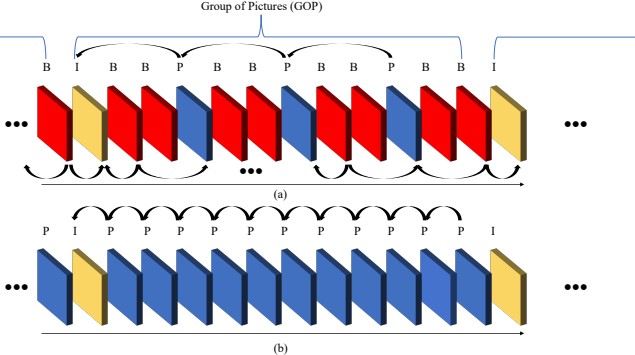

**Fig. 2.** (a) A modern GOP structure (I-B-B-P-B-B···), which is capable of getting a high amount of data compression. (b) An outdated GOP structure (I-P-P-P-P-P···), which consists of only P-frames and is not used today. Bold arrows denote reference dependencies, and thin arrows denote time flows (from left to right).

**Multi-Modal Pooling**  Pooling methods are required in two-stream networks [32,40] as well as other feature fusion models. [40] utilizes average pooling. [27] proposed bilinear pooling to model local parts of an object: two feature representations are learned separately and then multiplied using the outer product to obtain the holistic representation. [43] combines the two-stream network with a compact bilinear representation [12]. However, most of existing pooling models can combine only two features, while in compressed videos, more than two modalities exist. To address such a challenging problem, we propose a novel Aligned Temporal Trilinear Pooling module to exploit the complementary information contained in the three modalities extracted from compressed videos.

**Lightweight Neural Networks**  Recently, lightweight neural networks including SqeezeNet [16], Xception [4], ShuffleNet [28], MobileNet [15] and Efficient-Net [36] have been proposed, with the number of parameters and computational cost being reduced significantly. Since we focus on lightweight action recognition, they all can be used as the backbone network. In this work, EfficientNet [36] is selected. Note that a lightweight backbone typically leads to performance degradation in action recognition, and hence the temporal fusion and knowledge distillation model designs are introduced in our ATTP model.

## 3  Methodology

### 3.1  Modeling Compressed Representation

Modern compression codecs use motion compensation to convert successive raw video frames into several groups of pictures (GOPs), where each GOP contains one I-frame (Intra-coded frame), one or more P-frames (Predicted frames), and one or more B-frames (Bi-directional predicted frames). From Fig. 2 (a), we can see an example GOP frame pattern used by modern compression codecs.

I-frame ($\mathbf{I}$) is the first frame in the GOP, which is compressed with image compression codecs (e.g., JPEG). P-frames hold the changes in the images w.r.t. the preceding frame and thus save the storage space, while B-frames save even more space by using differences between the current frame and both the preceding and following frames to represent their content. Consequently, images from B-frames and P-frames are stored in a compressed format and are reconstructed using these encoded offsets, namely motion vectors ($\mathbf{MV}$) and residuals ($\mathbf{R}$). Let $x$, $y$, and $z$ denote the coordinates in the three dimensions. The relation between video frames, motion vectors, and residuals can be formalized as:

$$g_{\mathrm{I}}(x,y,z) = f_{\mathrm{I}}(x,y,z)$$
$$g_{\mathrm{P}}(x,y,z) = f_{\mathrm{P_{ref}}}(x + mv_{\mathrm{P}}(x,y,0), y + mv_{\mathrm{P}}(x,y,1), z) + r_{\mathrm{P}}(x,y,z) \qquad (1)$$
$$g_{\mathrm{B}}(x,y,z) = f_{\mathrm{B_{ref}}}(x + mv_{\mathrm{B}}(x,y,0), y + mv_{\mathrm{B}}(x,y,1), z) + r_{\mathrm{B}}(x,y,z),$$

where $g_{\mathrm{F}} \in \mathbb{R}^{h \times w \times 3}$ denotes the reconstructed image of the F-frame (F = I, P, or B) and $f_{\mathrm{F}} \in \mathbb{R}^{h \times w \times 3}$ denotes its counterpart in raw image, with $h$ and $w$ being respectively the height and width. Moreover, $mv_{\mathrm{P}}$ (or $mv_{\mathrm{B}}$) $\in \mathbb{R}^{h \times w \times 2}$ describes the predicted block-level motion trajectories from the reference frame to the current P-frame (or B-frame). For each position $(x, y)$ in $mv$, $mv(x, y, 0)$ and $mv(x, y, 1)$ depict the horizontal and vertical movements. In addition, $r_{\mathrm{P}}$ (or $r_{\mathrm{B}}$) $\in \mathbb{R}^{h \times w \times 3}$ is the RGB-like image describing the residual error between the original P-frame (or B-frame) and its predicted frame. $f_{\mathrm{P_{ref}}}$ and $f_{\mathrm{B_{ref}}}$ are frames referenced by P-frames and B-frames. $f_{\mathrm{P_{ref}}}$ denotes the previous I or P-frame of the current P-frame, and its motion vector contains only backward reference. Note that B-frame has both forward and backward motion information, and different codecs choose different $f_{\mathrm{P_{ref}}}$. Since both $\mathbf{MV}$ and $\mathbf{R}$ are available in compressed videos, we can readily extract them.

However, since B-frames require a higher computational cost to compress than P-frames, earlier codecs such as MPEG-4 [31] use only P-frames for low-cost applications by default, where P-frames substitute all B-frames since they are capable of getting the highest computational efficiency (see Fig. 2 (b)). Because of the complicated modeling of B-frames, existing compressed video action recognition works [45,30] extract motion vectors and residual representations from only P-frames and thus are only applicable to the outdated MPEG-4 standard, and Zhang et al. [49,50] simply treat B-frames as P-frames. All these methods only consider forward reference, and at least half of the temporal information is thus ignored. When applied to the modern video codecs such as H.264 [44] and HEVC [35], these methods cannot take advantage of both forward and backward motion information induced by B-frames, which thus limits their performance. Note that CoViAR [45] and DMC-Net [30] even need to re-encode the compressed videos using the MPEG-4 codec for avoiding handling B-frames, which leads to extra cost on both processing time and storage.

### 3.2  Lightweight Video Action Recognition with Multiple Modalities

We now formally define the lightweight video action recognition problem as follows. Given the three modalities (i.e., $\mathbf{I}$, $\mathbf{MV}$, and $\mathbf{R}$) extracted from a

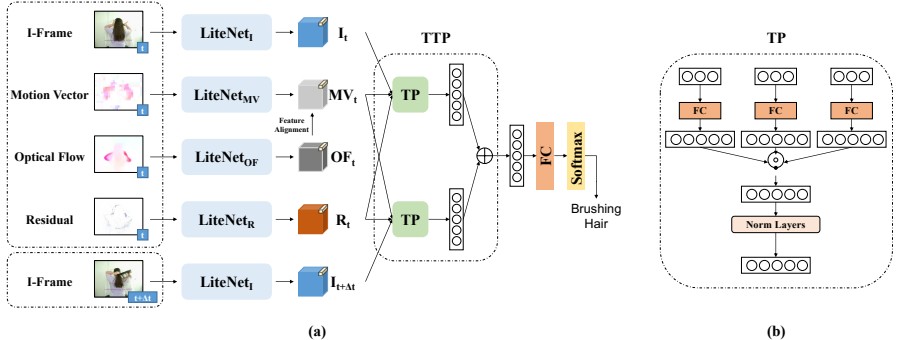

(a)                                                    (b)

**Fig. 3.** (a) Overview of our Aligned Temporal Trilinear Pooling (ATTP). Note that optical flow is only used at the training phase, but not at the test phase. (b) Complete design of the Trilinear Pooling (TP) module.

compressed video, our goal is to perform high-speed action recognition using a model with fewer parameters. Recent works utilize different CNNs to process the modalities independently. For example, in [45], ResNet-152 [13] is used to process **I**, which is effective but has extremely high computational cost and require large storage space (see Table 1). In this paper, we adopt a lightweight network LiteNet instead: $\text{LiteNet}_I$, $\text{LiteNet}_{MV}$, and $\text{LiteNet}_R$ take **I**, **MV**, and **R** as inputs, respectively. Each of them has the same EfficientNet [36] network architecture. Note that the three modalities are only fused at the test phase in [45], and by doing so, the interactions between different modalities are not fully explored during training for action recognition. More effective multi-modal fusion is thus needed. To that end, we propose our Aligned Temporal Trilinear Pooling (ATTP) framework (see Fig. 3(a)), which is detailed next.

### 3.3 Trilinear Pooling

We propose a novel trilinear pooling module to model three-factor variations together. This is motivated by bilinear pooling models [27,47], which are initially proposed to model two-factor variations, such as "style" and "content". For lightweight video action recognition, we generalize the bilinear pooling method to fuse the three modalities extracted from compressed videos.

Specifically, a feature vector is denoted as $x \in \mathbb{R}^c$, where $c$ is the dimensions of the feature $x$. The bilinear combination of two feature vectors with the same dimension $x \in \mathbb{R}^c$ and $y \in \mathbb{R}^c$ is defined as $xy^{\mathrm{T}} \in \mathbb{R}^{c \times c}$ [27]. In general, given two representation matrices $X = [x_1; x_2; \cdots; x_K] \in \mathbb{R}^{c \times K}$ and $Y = [y_1; y_2; \cdots; y_K] \in \mathbb{R}^{c \times K}$ for two frames, a pooling layer takes the following bilinear combination:

$$f_{\mathrm{BP}}(X, Y) = \frac{1}{K} \sum_{i=1}^{K} x_i y_i^{\mathrm{T}} = \frac{1}{K} XY^{\mathrm{T}}. \tag{2}$$

It can be seen clearly that the above bilinear pooling allows the outputs ($X$ and $Y$) of the feature extractor to be conditioned on each other by considering all

their pairwise interactions in the form of a quadratic kernel expansion. However, this results in very high-dimensional features with a large number of parameters involved. To address this problem, Multi-modal Factorized Bilinear (MFB) [47] introduces an efficient attention mechanism into the original bilinear pooling based on the Hadamard product. Let $D$ be the number of projection matrices. The MFB model w.r.t. projection $i$ ($i = 1, ..., D$) is defined as:

$$f_{\text{MFB}}(x, y)_i = x^{\text{T}} U_i V_i^{\text{T}} y = \mathbb{1}^{\text{T}}(U_i^{\text{T}} x \odot V_i^{\text{T}} y), \tag{3}$$

where $U_i \in \mathbb{R}^{c \times d}$ and $V_i \in \mathbb{R}^{c \times d}$ are projection matrices, $\odot$ is the Hadamard product, $\mathbb{1} \in \mathbb{R}^d$ is an all-one vector, and $d$ denotes the dimension of these factorized matrices. Therefore, we only need to learn $U = [U_1; U_2; \cdots; U_D] \in \mathbb{R}^{c \times d \times D}$ and $V = [V_1; V_2; \cdots; V_D] \in \mathbb{R}^{c \times d \times D}$.

Inspired by MFB, we propose a novel trilinear pooling method, which aims to fuse three feature vectors ($x$, $y$, and $z$). Unlike bilinear pooling that can combine only two feature vectors, our Trilinear Pooling method fuse $x$, $y$ and $z$ using the Hadamard product:

$$f_{\text{TP}}(x, y, z) = \mathbb{1}^{\text{T}}(U^{\text{T}} x \odot V^{\text{T}} y \odot W^{\text{T}} z), \tag{4}$$

where $W$ is also a projection matrix $W = [W_1; W_2; \cdots; W_D] \in \mathbb{R}^{c \times d \times D}$, and $f_{\text{TP}}$ denotes the output of trilinear pooling. Note that our trilinear pooling becomes MFB if all elements in $W$ and $z$ are fixed as 1. When the inputs are generalized to feature maps (i.e., $X = [x_i], Y = [y_i], Z = [z_i] \in \mathbb{R}^{c \times K}$), every position of these feature maps makes up one group of inputs, and the outputs of them are summed element-wised as follows:

$$f_{\text{TP}}(X, Y, Z) = \sum_{i=1}^{K} f_{\text{TP}}(x_i, y_i, z_i). \tag{5}$$

We thus utilize trilinear pooling to obtain the multi-modal representation of the $t$-th GOP by fusing I-frame $I_t$, motion vector $MV_t$ and residual $R_t$ (see Fig. 3(b)):

$$f_{\text{TP}}(I_t, MV_t, R_t) = \sum_{i=1}^{K} f_{\text{TP}}(I_{t,i}, MV_{t,i}, R_{t,i}), \tag{6}$$

where $I_t$, $MV_t$ and $R_t$ are the output feature maps from LiteNet$_I$, LiteNet$_{MV}$ and LiteNet$_R$, respectively. We set LiteNet as EfficientNet. For each GOP $t$, the I-frame is selected as $I_t$, while one $MV_t$ and one $R_t$ are randomly selected. As in [21], the trilinear vector is then processed with a signed square root step ($f \leftarrow \text{sign}(f)\sqrt{|f|}$), followed by $l_2$ normalization ($f \leftarrow f/||f||$).

### 3.4   Temporal Trilinear Pooling

Motion vector is initially introduced to represent the temporal structure as the optical flow does. However, compared to the high-resolution optical flow, motion vector is much coarser: Since it only describes the movement on macroblock-level (e.g., 16*16 pixels), all values within the same macroblock are identical. Although we have proposed to use trilinear pooling to address this drawback, the

temporal information still needs to be explicitly explored. We note that, because residuals represent the difference between frames, they are strongly correlated with motion vectors. Therefore, we propose to model the motion vectors and the residuals jointly. Note that the fusion of $I_t$, $MV_t$ and $R_t$ within only one GOP is not enough to capture the temporal information. We thus further choose to include the adjacent GOP's information. Specifically, in addition to calculating $f_{TP}(I_t, MV_t, R_t)$ by trilinear pooling, we also combine $MV_t$ and $R_t$ with $I_{t+\Delta t}$ (i.e., the I-frame in the adjacent GOP). The output of temporal trilinear pooling (TTP) is defined as (see Fig. 3(a)):

$$f_{TTP}(t) = f_{TP}(t, 0) + f_{TP}(t, \Delta t), \tag{7}$$

where $f_{TP}(t, \Delta t)$ denotes $f_{TP}(I_{t+\Delta t}, MV_t, R_t)$ for notation simplicity. In this paper, we sample the offset $\Delta t$ from $\{-1, 1\}$ during the training stage. During the test stage, $\Delta t$ is fixed as 1 for the first GOP and $-1$ for other GOPs. This temporal fusion method solves the temporal representation drawback without introducing extra parameters, which is efficient and lightweight. The TTP representation is further put into a fully connected layer to calculate the classification scores $\mathbf{s}(t) = P^T f_{TTP}(t)$, where $P \in \mathbb{R}^{D \times C}$ is learnable parameters and $C$ is the number of categories.

### 3.5  Feature Alignment

Since motion vectors can only be regarded as a blurred version of optical flow, we choose to boost them with optical flow by feature alignment based knowledge distillation. Specifically, we employ another lightweight network LiteNet$_{OF}$ which takes optical flow information as input, and align the features generated by LiteNet$_{MV}$ to those by LiteNet$_{OF}$, as illustrated in Fig. 3(a). Different from the original method [14] that transfers knowledge from complex models to simple models, our feature alignment sets LiteNet$_{MV}$ and LiteNet$_{OF}$ to be the same lightweight network. Notably, we find that our feature alignment performs better than conventional knowledge distillation based on classification probability alignment. Moreover, our feature alignment is also much more efficient in the training phase since we do not need a large CNN as a teacher.

Our feature alignment module tries to minimize the difference between features generated by LiteNet$_{MV}$ and those by LiteNet$_{OF}$. Note that Zhang et al. [49,50] utilized all layers for knowledge distillation from the teacher network to the student network, while we only exploit features before the fully-connected (FC) layer and those after the FC layer for feature alignment (which is more efficient and thus more suitable for lightweight action recognition). Let $f_{MV}^{(3d)}$ (or $f_{OF}^{(3d)}$) be features before the FC layer of LiteNet$_{MV}$ (or LiteNet$_{OF}$) and $f_{MV}^{(fc)}$ (or $f_{OF}^{(fc)}$) as features after the FC layer of LiteNet$_{MV}$ (or LiteNet$_{OF}$). The loss for feature alignment is given by:

$$\mathcal{L}_{Align} = ||f_{MV}^{(3d)} - f_{OF}^{(3d)}||^2 + ||f_{MV}^{(fc)} - f_{OF}^{(fc)}||^2. \tag{8}$$

Given that video action recognition is essentially a multi-class classification problem, we utilize the standard cross-entropy loss for training the TTP module:

$$\mathcal{L}_{TTP}(t) = -\log \text{softmax}(\mathbf{s}_{gt}(t)), \tag{9}$$

where $\mathbf{s}_{gt}(t)$ is the predicted score for $t$-th GOP with respect to its ground-truth class label. The total loss of our Aligned Temporal Trilinear Pooling (ATTP) model is defined as follows:

$$\mathcal{L}_{ATTP} = \mathcal{L}_{TTP} + \lambda \mathcal{L}_{Align}. \tag{10}$$

where $\lambda$ is the weight parameter (we empirically set $\lambda = 1$ in this work).

## 4    Experiments

### 4.1    Datasets and Settings

In this paper, the main results are reported on two widely-used benchmark datasets, namely **HMDB-51** [23] and **UCF-101** [33], as in [40,45,30] for direct comparison. HMDB-51 contains 6,766 videos from 51 action categories, while UCF-101 contains 13,320 videos from 101 action categories. Both benchmark datasets have three officially given training/test splits. In HMDB-51, each training/test split consists of 3,570 training clips and 1,530 testing clips. In UCF-101, each training/test split consists of approximately 9,600 clips in the training split and 3,700 clips in the test split. Since each video in these two datasets is a short clip belonging to a single category, we employ top-1 accuracy on video-level class predictions (for the test split of each dataset) as the evaluation metric.

As in [40,45], we resize frames in all videos to $340 \times 256$. In order to implement our proposed model on resource-limited devices, we choose the EfficientNet B1 [36] pre-trained on ImageNet as the core CNN module to extract the representations of I-frames, motion vectors, and residuals. All the parameters of the projection layers are randomly initialized.

### 4.2    Efficiency Test Results

We firstly demonstrate the per-frame running time and FPS of our model in both limited-resource and sufficient-resource environments. We make comparisons to the state-of-the-art CoViAR [45] and DMC-Net [30] since they also exploit compressed videos for action recognition. Note that CoViAR does not use optical flow, while DMC-Net uses optical flow but only during the training phase like our model. However, both CoViAR and DMC-Net utilize ResNet-152 for I-frames and two ResNet-18 for motion vectors and residuals independently, which means their models are much larger in size and slower to run.

We compare the efficiency of the three models (CoViAR, DMC-Net, and our ATTP model) under precisely the same test setting. On the Dell R730 platform, the preprocessing phase (including loading networks) is mainly run on two Intel Xeon Silver 4110 CPUs, and the CNN forwarding phase (including extracting

**Table 1.** Comparison of per-frame inference efficiency. Following CoViAR [45] and DMC-Net [30], we forward multiple CNNs concurrently. CoViAR and DMC-Net cannot run on Jetson TX2 due to out of memory (OOM).

| Method | Platform | Time(ms) | | FPS | |
|---|---|---|---|---|---|
| | | Preprocess | CNN | Preprocess | CNN |
| ATTP (ours) | Jetson | **12.2** | **24.6** | **82.1** | **40.7** |
| CoViAR | Jetson | OOM | OOM | OOM | OOM |
| DMC-Net | Jetson | OOM | OOM | OOM | OOM |
| ATTP (ours) | R730 | **0.6** | **4.3** | **1587.3** | **233.6** |
| CoViAR | R730 | 7.8 | 5.1 | 127.6 | 194.9 |
| DMC-Net | R730 | 7.8 | 7.0 | 127.6 | 142.9 |

motion vectors and residuals) is mainly run on one TITAN Xp GPU. As shown in Table 1, our ATTP model runs faster among three models in both preprocessing and CNN phases on the Dell R730 platform. The preprocessing time on contrast is particularly stark. This is due to that both CoViAR and DMC-Net have three large networks with lots of parameters, resulting in massive cost on loading the networks. For the efficiency test on the resource-limited device, the experiments are conducted on the Nvidia Jetson TX2 platform. The preprocessing phase runs on Dual-core Denver 2 64-bit CPU, and the CNN forward phase runs on the GPU. The results in Table 1 demonstrate that CoViAR and DMC-Net are too large to be employable on this device, while our ATTP framework fits well in the embedded environment and runs very fast. These results thus show clearly that our ATTP model outperforms the other two models on efficiency and is the only one that is suitable for embedded AI devices.

### 4.3   Comparative Results on Accuracy

Now we make a comprehensive comparison between our ATTP method and other state-of-the-art action recognition methods. To this end, we compute not only the top-1 accuracy but also the efficiency (i.e., parameters and GFLOPs) as the evaluation metrics for action recognition. In our experiments, the compared methods can be divided into two groups: 1) **Raw-Video Based Methods**: LRCN [6] and Composite LSTM [34] utilize RNNs to process the optical flow information, while Two-stream [32] adopts two-way CNNs to process the optical flow information. ResNet-152 [13], C3D [37], I3D (RGB-only) [2], P3D [29] and TSN (RGB-only) [40] employ large CNN models over RGB frames without using other information. ECO [51] and TSM [25] make efforts on lightweight video analysis models, but these models do not work on compressed videos. 2) **Compressed-Video Based Methods**: DTMV-CNN [50] integrates both compressed videos and raw ones into a single two-stream network, while CoViAR [45] and DMC-Net [30] are among the most closely related models (w.r.t. our ATTP model) that only exploit compressed videos for action recognition. In particular, DMC-Net, DTMV-CNN, and our ATTP model use optical flow during the training phase (but not during the test phase).

**Table 2.** Comparative results on the two benchmark datasets for both raw-video based methods and compressed-video based ones. "CV." denotes the usage of compressed videos for action recognition. "OF." denotes the usage of optical flow. ‡ indicates that the model only uses RGB frames.

| Model | Setting | | Efficiency | | Accuracy | |
|---|---|---|---|---|---|---|
| | CV. | OF. | Param.(M) | GFLOPs | HMDB | UCF |
| LRCN [6] | N | Y | 114.8 | 15.5 | – | 82.7 |
| Comp. LSTM [34] | N | Y | – | – | 44.0 | 84.3 |
| Two-Stream [32] | N | Y | 46.6 | 3.3 | 59.4 | 88.0 |
| ResNet-152 [13] | N | N | 60.2 | 11.3 | 48.9 | 83.4 |
| C3D [37] | N | N | 78.4 | 38.5 | 51.6 | 82.3 |
| I3D‡ [2] | N | N | 24 | 108 | 49.8 | 84.5 |
| P3D [29] | N | N | 98 | – | – | 88.6 |
| TSN‡ [40] | N | N | 28.2 | 4.3 | – | 85.7 |
| ECO (4 frames) [51] | N | N | 23.8 | 16 | 61.7 | 90.3 |
| TSM (Kinetics) [25] | N | N | 24.3 | 33 | **64.7** | **91.7** |
| DTMV-CNN [50] | Both | Y | 181.3 | 83.4 | 55.3 | 87.5 |
| CoViAR [45] | Y | N | 83.6 | 14.9 | 59.1 | 90.4 |
| DMC-Net [30] | Y | Y | 83.6 | 15.1 | 62.8 | 90.9 |
| ATTP (ours) | Y | Y | **23.4** | **3.0** | **62.9** | **91.1** |

The comparative results are shown in Table 2. We have the following observations: (1) Our ATTP model is the most efficient for video action recognition. Specifically, it contains only $23.4 \times 10^6$ parameters and has only average 3.0 GFLOPs over all frames. (2) Among all compressed video action recognition methods (including CoViAR and DMC-Net), our model performs the best w.r.t. both efficiency and accuracy. (3) As compared to the strongest baseline DMC-Net, although our model only achieves marginal improvement on accuracy, this is obtained with much fewer parameters and only one-tenth of GFLOPS. This is mainly due to its ability to more effectively fuse multiple modalities using the proposed trilinear pooling module. (4) As compared to CoViAR, our model saves nearly 70% of the storage with a clear advantage on accuracy. To better understand the reason why our lighter model can yield more accurate recognition, in our ablation study (to be presented next), it is noted that "I+MV+R" in Table 3 is essentially CoViAR by using EfficientNet as the core CNN module. Under such a fair comparison setting (same CNN backbone), our model consistently yields accuracy improvements over CoViAR on all dataset splits. (5) Our model achieves even higher (or comparable) accuracies w.r.t. recent raw-video based models (TSM uses the external Kinetics [20] for pre-training), showing its potential for directly analyzing compressed video for action recognition.

### 4.4   Ablation Study

**Ablation Study over Different Modalities** We conduct experiments to show the benefits of using our ATTP model compared with single modality and other fusion options. Specifically, we uniformly use three EfficientNet networks to process the three components (**I**, **MV**, and **R**) extracted from compressed videos

**Table 3.** Ablative results (%) for our ATTP model using different modalities on the two benchmarks (each has three splits).

| Dataset | HMDB-51 | | | | UCF-101 | | | |
|---|---|---|---|---|---|---|---|---|
| Splits | S1 | S2 | S3 | Avg. | S1 | S2 | S3 | Avg. |
| I | 51.6 | 51.0 | 52.0 | 51.5 | 84.0 | 83.0 | 84.5 | 83.8 |
| MV | 45.2 | 44.8 | 44.9 | 45.0 | 70.1 | 70.5 | 73.2 | 71.3 |
| R | 48.2 | 45.6 | 48.5 | 47.4 | 81.7 | 81.1 | 82.0 | 81.6 |
| I+MV+R | 60.5 | 57.1 | 57.9 | 58.5 | 86.1 | 86.3 | 87.3 | 86.6 |
| BP | 60.9 | 57.7 | 58.4 | 59.0 | 87.3 | 87.1 | 88.4 | 87.6 |
| TP | 61.6 | 58.5 | 59.4 | 59.8 | 87.7 | 87.6 | 89.0 | 88.1 |
| TTP | 62.1 | 59.0 | 59.9 | 60.3 | 88.3 | 88.2 | 89.5 | 88.7 |
| ATTP | **64.3** | **61.8** | **62.6** | **62.9** | **91.0** | **90.9** | **91.5** | **91.1** |

and demonstrate all the ablative results on the two benchmarks by training with different parts of our ATTP model. For single-modality based models, "I", "MV" and "R" denote the results obtained by using LiteNet$_I$, LiteNet$_{MV}$ and LiteNet$_R$, respectively. For the late-fusion based model, "I+MV+R" indicates that the output is fused by simply adding the score of the three CNNs together. We also compare our ATTP model with existing bilinear pooling models, which are the simplified version of our Trilinear Pooling. Since existing bilinear pooling methods cannot be directly adapted to the three modalities, we readily apply pairwise combination over them, and sum the three predicted classification scores together like "I+MV+R". Note that conventional bilinear pooling [27] and factorized bilinear pooling [21] have too many parameters to be efficient, we resort to compact bilinear pooling [12] (denoted as "BP") with much fewer parameters. Finally, "TP" denotes our Trilinear Pooling, "TTP" denotes our Temporal Trilinear Pooling, and "ATTP" denotes our Aligned Temporal Trilinear Pooling.

As shown in Table 3, single-modality based models (i.e., I, MV, or R) could not achieve good results without using multi-modal information, indicating that the compressed video needs to be fully explored to obtain high accuracy. I and R yield similar results because they both contain the RGB data: the I-frames contain a small number of informative frames, while the residuals contain a large number of less informative frames. Since the motion vectors only contain the motion information, MV could not perform as well as the other two. Moreover, for multi-modal fusion, all bilinear/trilinear pooling methods outperform I+MV+R, showing the power of pooling methods instead of linearly late fusion. Particularly, our TP method yields %1 gains over BP, validating the effectiveness of our pooling method. In addition, the improvements achieved by TTP over TP (and those achieved by ATTP over TTP) clearly show the importance of boosting motion vectors by inducing temporal context (and feature alignment).

**Ablation Study over Different Types of Frames** To show the contribution of each type of frames to action recognition, we make comparison among four versions of the late fusion model: (1) I – only I-frames are used to obtain the single modality **I**; (2) I+P – both I-frames and P-frames are used to extract the three modalities (i.e., **I**, **MV**, and **R**); (3) I+B – both I-frames and B-frames are used to extract the three modalities; (4) I+P+B – all three types of frames

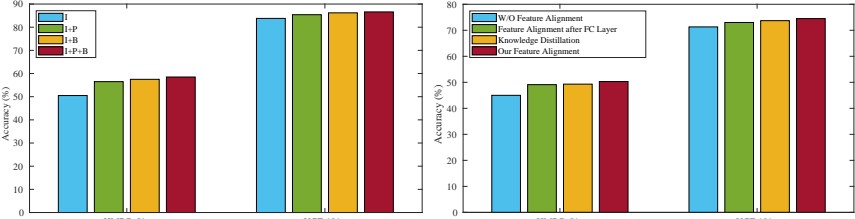

**Fig. 4.** Left: Ablative results (%) obtained by exploiting different types of frames for late fusion on the two benchmarks. Right: Ablative results (%) for our proposed feature alignment method on the two benchmarks. Only the single modality **MV** is used.

are used to extract the three modalities. The ablative results in Fig. 4 (left) show that: (a) The performance of action recognition continuously increases when more types of frames are added, validating the contribution of each type of frames. (b) The improvements achieved by I+B over I+P verify that B-frames are more important than P-frames for action recognition.

**Ablation Study for Feature Alignment**  To conduct the ablation study for our proposed feature alignment method, we make comparisons among four related methods: (1) W/O Feature Alignment – features are directly extracted from motion vectors, without feature alignment; (2) Feature Alignment after FC Layer – our proposed feature alignment method is used, but only the features after the FC layer are aligned; (3) Knowledge Distillation – the knowledge distillation [14] method is used for feature alignment; (4) Our Feature Alignment – our proposed feature alignment method defined in Eq. (8). The ablative results are presented in Fig. 4 (right). It can be seen that: (1) The three feature alignment methods outperform the 'W/O Feature Alignment' method, validating the effectiveness of feature alignment. (2) Our proposed feature alignment method performs the best among the three feature alignment methods. This suggests that aligning the features before and after the FC layer is more effective than feature alignment after the FC Layer and even than knowledge distillation.

## 5   Conclusion

In this work, we address a key limitation of existing deep neural networks-based video action recognition methods. That is, they either only work with raw RGB video frames instead of the compressed video directly, or heavy in model size and slow to run. We therefore proposed to address a more challenging task, namely lightweight compressed video action recognition. By employing EfficientNet as the backbone, we proposed a novel ATTP model to fuse the multiple modalities for lightweight video action recognition. Importantly, for the first time, our ATTP models the B-frames correctly, therefore being compatible with a wider range of contemporary codecs. The comparative results on three benchmark datasets show that our ATTP model outperforms the state-of-the-art alternatives in both efficiency and accuracy.

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
