# OpenReview forum: "Lightweight Action Recognition in Compressed Videos"
_thecvf.com/ECCV/2020/Workshop/VIPriors — VIPriors Oral_

### Official Review · AnonReviewer2 · 2020-07-27
**Lightweight Action Recognition in Compressed Videos**

**Confidence:** 4
**Rating:** 8

**Review:**

#### 1. [Summary] In 2-3 sentences, describe the key ideas, experiments, and their significance.
The paper proposes a lightweight action recognition model which can run on embedded AI devices with compressed videos.

#### 2. [Strengths] What are the strengths of the paper? Clearly explain why these aspects of the paper are valuable.
- Lightweight model and can fit embedded AI device
- Speed, efficiency and accuracy
- Extensive related works and ablation study

#### 3. [Weaknesses] What are the weaknesses of the paper? Clearly explain why these aspects of the paper are weak.
- Although some of information are refered as "in the suppl. material", nothing is sent as supplamentary materials.

#### 4. [Overall rating] Paper rating
8

#### 5. [Justification of rating] Please explain how the strengths and weaknesses aforementioned were weighed in for the rating.


#### 6. [Detailed comments] Additional comments regarding the paper (e.g. typos or other possible improvements you would like to see for the camera-ready version of the paper, if any.)

- Can the proposed method work with raw videos as well?
- L.429: Are all the baselines use the same spatial size of the proposed method?
- L.600: The improvement of B-frames is not significant. Is there any other evidence to show B-frames are more important than P-frames?
- What are the limitations of the work?
Typos:
- L.154: compressed
- L.175: utilizes
- L.178: combines
- L.438: giving full form of FPS

---

### Official Review · AnonReviewer1 · 2020-07-27
**Lightweight Action Recognition in Compressed Videos**

**Confidence:** 3
**Rating:** 8

**Review:**

1. [Summary] In 2-3 sentences, describe the key ideas, experiments, and their significance.

The paper describes a new lighter and more efficient model (ATTP) for the problem of Compressed Video Action Recognition. As backbone, the adopt the EfficientNet network. To better use the information of frames, motion and residuals, the authors also introduce a temporal fusion module (Trilinear Pooling) and a feature alignment method. Experiments show ATTP can reach state-of-the-art results by using a lighter model. Ablation experiments also demonstrate the benefits of all the modules.

2. [Strengths] What are the strengths of the paper? Clearly explain why these aspects of the paper are valuable.

The whole paper is well written and the motivations are well stated.
The set of experiments is very readable and clear.
The authors clearly demonstrate they can achieve state-of-the-art results using a lighter model.

3. [Weaknesses] What are the weaknesses of the paper? Clearly explain why these aspects of the paper are weak.

More than clear weaknesses, I miss some little further justifications:
-	It is clear that feature alignment improves accuracy but interestingly the lambda parameter of the whole ATTP loss (Eq. 10) is set to 1. I would also have appreciated an experiment varying this parameter to see the actual influence of it.
-	Regarding the type of frames used, although Fig. 4 seems pretty informative, it would also be interesting to see how I, B and P frames affect to computation efficiency, with an experiment like in Table 1.

4. [Overall rating] Paper rating.

8

5. [Justification of rating] Please explain how the strengths and weaknesses aforementioned were weighed in for the rating.

Aforementioned strengths and weaknesses explain the rating, but to clarify: good readability, well motivated, sufficient set of experiments and results.

6. [Detailed comments] Additional comments regarding the paper (e.g. typos or other possible improvements you would like to see for the camera-ready version of the paper, if any.)

The Action Localization experiment that authors indicated to be in the supplementary material seems interesting. I wonder if authors plan to release the supplementary material as well.

---

### Decision · Program_Chairs · 2020-07-29

**Decision:**

Accept (Oral)

**Comment:**

It is our pleasure to inform you that your paper has been accepted to the oral track of the 1st Visual Inductive Priors for Data-Efficient Deep Learning Workshop.

Please note the following deadlines:
* August 11, 2020 - workshop material, including:
 * paper in PDF format;
 * pre-recorded video presentation;
 * slides of the presentation in PDF.
* September 15, 2020 - camera-ready paper

The reviews can be found on OpenReview. Please take these comments and suggestions into account when preparing the camera-ready version of your paper, which is due September 15, 2020. The camera-ready paper should be uploaded to OpenReview.

As part of the workshop, each paper for oral presentation must submit a pre-recorded 5 minute talk before August 11, 2020. You will receive more information on how to upload the material shortly. The requirements for the video are:
* Duration: maximum 5 minutes
* MP4 format
* File size max. 100 MB
* Has an inset with a video of the speaker
* 16:9 aspect ratio (strongly preferred)
* 1920x1080 resolution (strongly preferred, at least 720 height)

Our suggested software for pre-recording your presentation is Zoom. For more information, please refer to the following guides:
How to record with Zoom Guide: http://homepages.inf.ed.ac.uk/rbf/ECCV2020HowtoRecordusingZoom.pdf
How to Record with Zoom tutorial: https://www.youtube.com/watch?v=CR199W7HdC0
Please ensure that at least one of the authors of the paper is available to attend the workshop during the allotted times. Note that the workshop will take place in two sessions spread across time zones (details are to follow). We will send instructions on how to connect to the workshop as soon as possible. The schedule for all talks and papers will be posted soon at the workshop website: https://vipriors.github.io.

We look forward to seeing you at the workshop!